

# A review of sebum in mammals in relation to skin diseases, skin function, and the skin microbiome

Karen Vanderwolf[1], Christopher Kyle[2,3] and Christina Davy[1,4]

[1] Department of Environmental and Life Sciences, Trent University, Peterborough, Ontario, Canada
[2] Forensic Science Department, Trent University, Peterborough, Ontario, Canada
[3] Natural Resources DNA Profiling and Forensics Center, Trent University, Peterborough, Ontario, Canada
[4] Department of Biology, Carleton University, Ottawa, Ontario, Canada

## ABSTRACT

Diseases vary among and within species but the causes of this variation can be unclear. Immune responses are an important driver of disease variation, but mechanisms on how the body resists pathogen establishment before activation of immune responses are understudied. Skin surfaces of mammals are the first line of defense against abiotic stressors and pathogens, and skin attributes such as pH, microbiomes, and lipids influence disease outcomes. Sebaceous glands produce sebum composed of multiple types of lipids with species-specific compositions. Sebum affects skin barrier function by contributing to minimizing water loss, supporting thermoregulation, protecting against pathogens, and preventing UV-induced damage. Sebum also affects skin microbiome composition both via its antimicrobial properties, and by providing potential nutrient sources. Intra- and interspecific variation in sebum composition influences skin disease outcomes in humans and domestic mammal species but is not well-characterized in wildlife. We synthesized knowledge on sebum function in mammals in relation to skin diseases and the skin microbiome. We found that sebum composition was described for only 29 live, wild mammalian species. Sebum is important in dermatophilosis, various forms of dermatitis, demodicosis, and potentially white-nose syndrome. Sebum composition likely affects disease susceptibility, as lipid components can have antimicrobial functions against specific pathogens. It is unclear why sebum composition is species-specific, but both phylogeny and environmental effects may drive differences. Our review illustrates the role of mammal sebum function and influence on skin microbes in the context of skin diseases, providing a baseline for future studies to elucidate mechanisms of disease resistance beyond immune responses.

## INTRODUCTION

Understanding how some species or populations resist disease can inform management strategies, yet the underlying mechanisms leading to varied disease outcomes are poorly understood. Immune functions are an important driver of variation in responses to disease, but mechanisms on how the body resists pathogen entry and establishment before immune

Corresponding author
Karen Vanderwolf,
kjvanderw@gmail.com

responses are activated are understudied. Pathogens can enter the body through mucosal surfaces in the gastrointestinal, urogenital, and respiratory tracts (*Van Ginkel, Nguyen & McGhee, 2000*), as well as the skin. The skin surface of mammals is the main interface with the external environment, and the initial physical and chemical barrier to pathogens. Differences in this barrier among species and individuals may partially explain differences in disease susceptibility.

Skin is an effective barrier to the outside environment, nevertheless a variety of skin diseases occur in mammals caused by bacteria, fungi, viruses, environmental stressors (UV damage, chemical exposure), and invertebrate parasites (*Simpson et al., 2013*; *Lorch et al., 2015*; *Goodnight, 2015*; *Fountain et al., 2019*; *Fountain et al., 2017*; *Akdesir et al., 2018*; *Chuma et al., 2018*; *Le Barzic et al., 2021*; *Beckmen et al., 1997*; *Martinez-Levasseur et al., 2011*; *Kiula et al., 2021*; *Muneza et al., 2016*; *Doneley & Sprohnle-Barrera, 2021*; *Bressem et al., 2009*; *Munday, Whittington & Stewart, 1998*). Skin infections can compromise skin defenses thereby increasing susceptibility to other diseases (*Fitzgerald, Cooley & Cosgrove, 2008*). Skin diseases have resulted in significant population declines and localized extinctions in several mammalian species (*Zaria, 1993*; *Pence & Ueckermann, 2002*; *Dagleish et al., 2007*; *Cypher et al., 2017*; *Cheng et al., 2021*; *Escobar et al., 2021*). For example, white-nose syndrome, a fungal skin disease, has killed millions of bats of multiple species in the last 15 years, and some species are now listed as endangered in North America due to the effects of the disease (*Cheng et al., 2021*). New skin diseases continue to be discovered, such as the recent appearance of a skin disease of unknown etiology in numerous giraffe (*Giraffa camelopardalis*) populations across Africa (*Muneza et al., 2016*). These examples illustrate potential impacts of skin diseases on biodiversity.

Skin diseases of wild mammals vary among and within species in both occurrence and severity, but mechanisms influencing this variation are not fully understood (*Ringwaldt et al., 2021*; *Langwig et al., 2016*; *Akdesir et al., 2018*; *Escobar et al., 2021*; *Nimmervoll et al., 2013*; *Pence & Ueckermann, 2002*; *Oleaga et al., 2012*). Potential mechanisms include variation in host immune responses, pathogen lineage, host behavior, abiotic factors, skin microbiome, and skin physiology (*Nimmervoll et al., 2013*; *Moore et al., 2018*; *Davy et al., 2020*; *Turchetto et al., 2020*; *Vanderwolf et al., 2021*). Not all these mechanisms are conducive to management actions but clarifying the role of skin physiology in skin disease origin and progression may lead to effective treatments. Such information is particularly relevant for captive and endangered mammals in zoos (*Conde et al., 2011*) that can develop chronic and sometimes lethal skin diseases despite provision of treatment and supportive care (*Fountain et al., 2017*; *Dunn, Buck & Spotte, 1984*; *Bauwens, De Vroey & Meurichy, 1996*; *Nutting & Dailey, 1980*; *Takle et al., 2010*; *James & Raphael, 2000*; *Kloft, Ramsay & Sula, 2019*; *Hubbard, Schmidt & Fletcher, 1983*; *Montali et al., 1981*; *Brack et al., 1997*; *Muneza et al., 2016*; *Pollock, Rohrbach & Ramsay, 2000*; *Diniz, Costa & Oliveira, 1995*; *Munson et al., 1998*). Not all skin diseases are influenced by skin attributes, as pathogens can elude the skin barrier by entering the body through mucosal surfaces, insect bites, or skin trauma such as lumpy skin disease in wild and domestic bovines and Tasmanian devil facial tumour disease (*Cunningham et al., 2021*; *Namazi & Tafti, 2021*). Nevertheless, skin characteristics play an important role in susceptibility to a range of diseases.

Skin surface defense against microbial invasion includes the combined effects of epidermal desquamation, acidic pH, nutrient and water limitations, commensal microbes, antimicrobial lipids, antimicrobial peptides, and antibodies (*Harder, Schröder & Gläser, 2013*; *Naik et al., 2012*). One source of antimicrobial skin lipids are sebaceous glands in the skin that produce sebum, a substance composed of cell debris and nonpolar (neutral) lipids that coats the epidermis and hair or fur (*Smith & Thiboutot, 2008*; *Harder, Schröder & Gläser, 2013*). Sebum composition and quantity affects the composition and abundance of the skin microbiome (*Drake et al., 2008*; *Roux, Oddos & Stamatas, 2021*). Altered sebum composition and quantity are associated with human skin diseases, such as sebaceous gland hyperplasia in acne and hypoplasia in atopic dermatitis (*Zouboulis et al., 2008*; *Shi et al., 2015*; *Knox & O'Boyle, 2021*), and may also play a role in wildlife diseases. If sebum composition is altered this can affect sebum function, which in turn may impact disease establishment and progression (*Zouboulis et al., 2008*; *Desbois & Smith, 2010*; *Lovászi et al., 2018*). Previous reviews focused on sebum composition and biochemistry in mammalian species, but did not address skin diseases or sebum function (*Stewart & Downing, 1991*; *Nikkari, 1974*).

Our objective is to synthesize existing literature concerning sebum function in mammals as it relates to skin diseases and the skin microbiome to identify key knowledge gaps for future research. While we restricted the scope of this review to sebum, we acknowledge that interactions between sebum and epidermal lipids are likely also involved in the maintenance of healthy skin and in susceptibility to skin diseases. Most knowledge about sebum derives from human and domestic and laboratory mammal studies, so we also draw on these studies as they are likely applicable to wild mammals. We explore the following topics: (1) sebum function in mammals, (2) mammalian skin diseases associated with sebaceous glands, (3) factors influencing sebum composition and quantity among mammals, and (4) directions for future research on sebum in wild mammals (Fig. 1). Our review promotes a greater understanding of the role of sebum in emerging diseases and interspecific differences among wild mammals that is useful for researchers interested in skin health, including microbial assemblages, diseases, and physiology.

## Methods: database search and literature screening

To identify relevant literature, we searched Web of Science and Google Scholar using the search string: ((sebum OR sebaceous) AND (composition OR composed OR function OR epidermis OR epidermal OR skin OR epidemiology OR disease OR fungi) AND (bat OR wildlife OR mammal)). The exclusion phrase (-human -children) was included in the Google Scholar search to exclude acne literature and to focus on wild mammals. We ordered the Google Scholar search results by relevance. We retained peer-reviewed publications that described sebum composition or sebum function in relation to skin diseases of any wild mammalian species. We summarized studies on the sebum composition of wild mammals in Table 1. Since literature on sebum function in relation to skin diseases in wild mammals is sparse, we include studies on function from humans and domestic and laboratory mammals. We excluded articles about skin treatments in humans and domestic or laboratory animals, or histology (physical structure of skin, hair, and glands) to focus on sebum function in

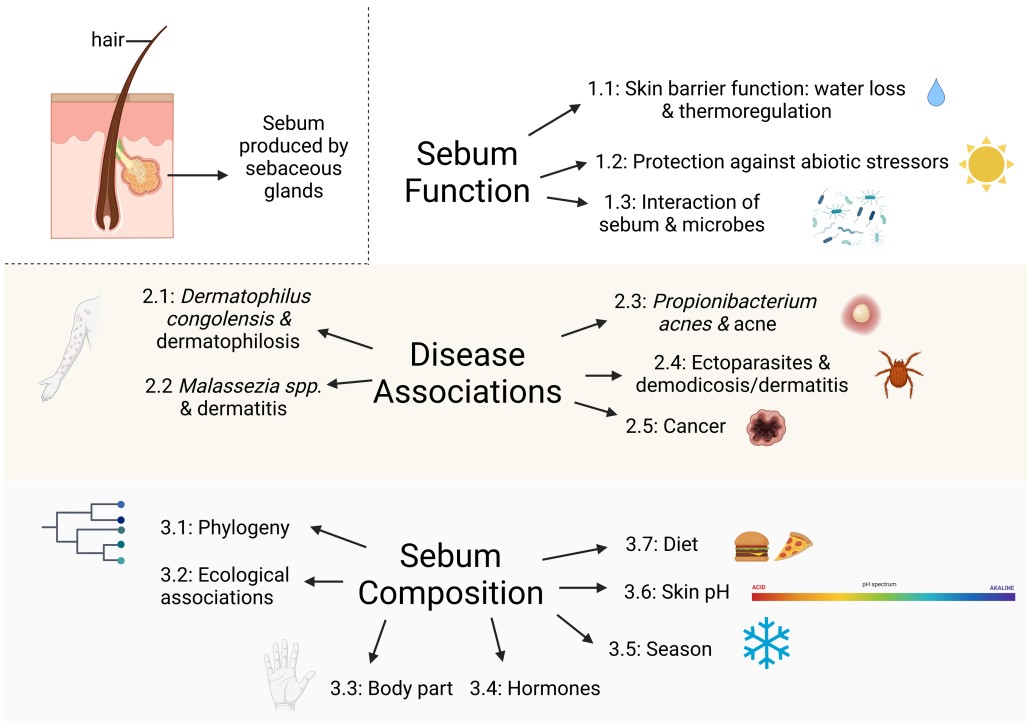

**Figure 1** Major topics and subtopics covered in our review comprising: (1) sebum function in mammals, (2) mammalian skin diseases associated with sebaceous glands, and (3) factors influencing sebum composition and quantity among mammals. Created with BioRender.com.

relation to skin disease (Fig. 2). We also excluded studies on the composition of scent glands from Table 1 to maintain the focus of this review on non-specialized sebaceous glands, although we briefly discuss scent gland functions in relation to non-specialized sebaceous glands. Scent glands can contain secretions from multiple sources, including sebaceous glands, apocrine (sweat) glands, urine, feces, and saliva, and often contain pheromones and other substances that are not present in non-specialized sebaceous glands over the rest of the skin (*Adams, Li & Wilkinson, 2018*; *Buesching, Newman & Macdonald, 2002*; *Buesching, Waterhouse & Macdonald, 2002*; *Dingzhen et al., 2006*; *Faulkes et al., 2019*; *Gassett et al., 1996*; *Jenkinson, Blackburn & Proudfoot, 1967*; *Kannan & Archunan, 1999*; *Khazanehdari, Buglass & Waterhouse, 1996*; *Martín et al., 2014*; *Muñoz Romo et al., 2012*; *Osborn et al., 2000*; *Salamon, Davies & Stoddart, 1999*; *Salamon & Davies, 1998*; *Sergiel et al., 2017*; *Waterhouse et al., 1996*). Studies generally report the composition of scent glands without differentiating which compounds originate from which source, consequently functions performed by scent glands cannot be specifically attributed to sebum. We included a total of 287 articles in our final review, the results of which are described below.

Peer

**Table 1** **Summary of available literature describing the sebum composition of live, wild mammals.** When more than one species of mammal was studied within a citation, we separated each species record, so some papers are represented more than once. Studies with an asterisk (*) also contain data on domestic or laboratory mammals. A study that analyzed the lipid composition of fur from dead mammals (road kill and skins in collections) was excluded (*Lindholm et al., 1981*), as lipid composition likely changes after death.

| Reference | Country | Months Samples Collected | Captivity Status | Species | Common Name | Sex | Age | n | Sample Type | Sebum or Epidermal lipids |
|---|---|---|---|---|---|---|---|---|---|---|
| *Frank et al. (2016)* | United States | Winter | Free-ranging | *Eptesicus fuscus* | Big brown bat | Unknown | Adult | 6 | Skin biopsy | Both |
| | | Winter, October, March | | *Myotis lucifugus* | Little brown myotis | | | 25 | | |
| | | Unknown | | *Eptesicus fuscus* | Big brown bat | | | | | |
| *Pannkuk et al. (2012)* | United States | June, July | Free-ranging | *Lasiurus borealis* | Eastern red bat | Both sexes | Adult | 5 pooled samples from 10-15 individuals for each spp.; 4 fur | Fur, wing-skin biopsy, scrubbing skin with cotton balls | Both |
| | | | | *Nycticeius humeralis* | Evening bat | | | | | |
| *Pannkuk et al. (2015)* | Canada | Winter | Captive | *Myotis lucifugus* | Little brown myotis | Unknown | Adult | 6 | Skin biopsy | Both |
| *Pannkuk et al. (2014a)* | United States | July | Free-ranging | *Lasiurus borealis* | Eastern red bat | Both sexes | Adult | Samples from *Pannkuk et al. (2012)* | hair clipped & lipids extracted | Sebum |
| *Pannkuk et al. (2013)* | United States | Unknown | Free-ranging | *Eptesicus fuscus* | Big brown bat | Both sexes | Adults | 5 pooled samples from 10-15 individuals | hair, wing surface, wing tissue | Both |
| | | | | *Lasiurus borealis* | Eastern red bat | | | | | |
| | | | | *Nycticeius humeralis* | Evening bat | | | | | |
| *Pannkuk et al. (2014b)* | United States | Unknown | Free-ranging | *Lasiurus borealis* | Eastern red bat | Both sexes | Adults & juveniles | 10 adults, 10 juveniles | Sebutape adhesive patches pressed to skin | Sebum |
| | | | | *Lasiurus cinereus* | Hoary bat | | | 6 | | |
| | | | | *Eptesicus fuscus* | Big brown bat | | | 12 | | |
| | | | | *Nycticeius humeralis* | Evening bat | | | 17 | | |
| | | | | *Myotis lucifugus* | Little brown myotis | | | 5 | | |
| | | | | *Myotis austroriparius* | Southeastern myotis | | | 11 | | |
| | | | | *Myotis septentrionalis* | Northern long-ear bat | | | 11 | | |
| | | | | *Myotis leibii* | Small-footed bat | | | 16 | | |
| | | | | *Myotis grisescens* | Gray bat | | | 10 | | |
| | | | | *Lasionycteris noctivagans* | Silver-haired bat | Both sexes | Adult | 6 | | |
| | | | | *Perimyotis subflavus* | Tricolored bat | | | 9 | | |
| | | | | *Corynorhinus rafinesquii* | Rafinesque's big-eared bat | | | 11 | | |
| | | | | *Corynorhinus townsendii ingens* | Ozark big-eared bat | | | 5 | | |

**Table 1** (*continued*)

| Reference | Country | Months Samples Collected | Captivity Status | Species | Common Name | Sex | Age | n | Sample Type | Sebum or Epidermal lipids |
|---|---|---|---|---|---|---|---|---|---|---|
| *Řezanka et al. (2015)* | Czech Republic | spring | Free-ranging | *Myotis myotis* | Greater mouse-eared bat | Male, Female | Adult | 6 Male, 6 Female | lipids isolated with chloroform from clipped fur | Sebum |
| *Downing & Stewart (1987)* | United States | April | Free-ranging | *Scalopus aquaticus* | Eastern mole | Unknown | Unknown | 1 | Body dipped in acetone | Sebum |
| *Wertz, Colton & Downing (1983)* * | Unknown | Unknown | Unknown | *Equus przewalskii* | Przewalski's horse | Unknown | Unknown | Unknown | Acetone poured on skin & then scraped off | Sebum |
| | | | | *Equus grevyi* | Grevy's zebra | | | | | |
| | | | | *Equus hemionus* | Onager | | | | | |
| *Roze, Locke & Vatakis (1990)* | United States | August, January, February | Free-ranging | *Erethizon dorsatum* | Porcupine | Unknown | Unknown | 7 | Quills | Sebum |
| *Wix, Wertz & Downing (1987)* * | United States | Unknown | Unknown | *Erethizon dorsatum* | Porcupine | Unknown | Unknown | 1 | hair, quills | Sebum |
| | | | | *Macaca fascicularis* | Crab-eating macaque | | | 1 | hair | |
| *Nishimaki-Mogami et al. (1988)* | Japan | Unknown | Unknown | *Macaca fascicularis* | Crab-eating macaque | Male | Unknown | 3 | Shaved skin wiped with acetone | Sebum |
| *Birkby, Wertz & Downing (1982)* * | Unknown | Unknown | Unknown | *Procyon lotor* | Racoon | Unknown | Unknown | 1 | Hair | Sebum |
| | | | | *Macaca fascicularis* | Crab-eating macaque | | | 1 | | |
| *Nicolaides, Fu & Rice (1968)* * | United States | Unknown | Captive | *Pan troglodytes* | Chimpanzee | Unknown | Unknown | 1 | hair clipped, skin washed with hexane | Sebum |
| | | | | species unknown | Baboon | | | 1 | | |
| *Gassett, Wiesler & Baker (1997)* | United States | December | Captive | *Odocoileus virginianus* | White-tailed deer | Male | 1.5-11.5 years | 10 | hair | Sebum |
| *Colton et al. (1986)* | United States | Unknown | Unknown | *Neogale vison* | Mink | Female | Unknown | 2 | Acetone poured over mid-section | Sebum |
| *Waldorf & Vedros (1978)* | United States | August | Free-ranging | *Callorhinus ursinus* | Northern fur seal | Male | Adult | 8 | acetone poured on skin | Sebum |

**Table 1** (*continued*)

| Reference | Country | Months Samples Collected | Captivity Status | Species | Common Name | Sex | Age | n | Sample Type | Sebum or Epidermal lipids |
|---|---|---|---|---|---|---|---|---|---|---|
| *Williams et al. (1992)* | United States | Unknown | Unknown | *Enhydra lutris* | California sea otter | Unknown | Adult | 1 | fur, skin biopsy | Both |
| *Davis et al. (1988)* | United States | Summer | Captive | *Enhydra lutris* | California sea otter | Male | Unknown | 8 | Fur | Sebum |
| *Lindholm & Downing (1980)* | United States | Unknown | Captive | *Lutra canadensis* | Otter | Unknown | Unknown | 1 | Fur | Sebum |
| | | | | *Castor canadensis* | Beaver | | | 1 | | |
| | | | | *Potos flavus* | Kinkajou | | | 2 | | |

| Reference | Method | Body Region | Conclusion |
|---|---|---|---|
| *Frank et al. (2016)* | GC-FAME | Wings | fatty acid composition differed between *Eptesicus fuscus* & *Myotis lucifugus* as the latter had more stearic acid & less palmitoleic, myristic, & oleic acid than the former. |
| | | | wing lipids of *M. lucifugus* prior to hibernation had higher myristic, stearic, & linoleic acid levels than during late hibernation |
| *Pannkuk et al. (2012)* | TLC, MALDI-TOF MS | Wings and back | Triacylglycerol proportions were higher in hair versus wing tissue. Triacylglycerol profiles were different among the 3 bat species. Bats had greater amounts of cholesterol & less squalene than humans |
| *Pannkuk et al. (2015)* | LC-MS | Wings | Wing tissue from bats with white-nose syndrome had different glycerophospholipid class composition from healthy tissue |
| *Pannkuk et al. (2014a)* | LC-MS | Unknown | targeted glycerophospholipids, found 152 types |
| *Pannkuk et al. (2013)* | GC-MS | Wings and back | bat sebum has a higher proportion of sterols & FFA compared to human sebum, & a lower proportion of squalene & monoacylglycerides |
| | | | red bats have lower ratios of FFA compared to big brown bats & evening bats |
| | | | Broad lipid classes did not differ between hair & wing, but ratios of specific FFA, monoacylglycerides, squalene, & sterol differed |
| *Pannkuk et al. (2014b)* | GC-FAME | Dorsal plagiopatagium | The ratios of fatty acid methyl acid types differed between *Lasiurus borealis* adults & juveniles |
| | | | characterized FAME which were similar among species |
| *Řezanka et al. (2015)* | Direct infusion MS | Unknown | composition of wax esters but not steryl esters differed between sexes |
| *Downing & Stewart (1987)* | TLC | Full body up to neck | skin lipids consisted of squalene, wax esters, & sterol esters with small amounts of triglycerides, FFA, free sterols, free fatty alcohols |
| *Wertz, Colton & Downing (1983)* * | TLC | Unknown | skin lipids consisted of cholesterol, cholesteryl esters, lactones, wax diesters |
| *Roze, Locke & Vatakis (1990)* | GC-FAME | Back and tail | Lipids on quills had more FFA in August compared to January/February; FFA inhibited growth of 6/10 bacteria strains tested. |
| *Wix, Wertz & Downing (1987)* * | TLC | Unknown / Unknown | lipids consisted of ceramides, cholesteryl sufate, glycosylceramides |
| *Nishimaki-Mogami et al. (1988)* | TLC | Back | lipids consisted of sterol esters, cholesterol, wax diesters |
| *Birkby, Wertz & Downing (1982)* * | TLC | Unknown | contain ceramides, polar glycolipids, cholesterol, FFA |
| | | | contain ceramides, polar glycolipids, cholesterol, FFA |
| | | | contain ceramides, polar glycolipids, phospholipids |
| *Nicolaides, Fu & Rice (1968)* * | TLC | back | lipid composition differs among species in the presence of squalene, sterol esters, wax esters, triglycerides, FFA, & diesters |
| *Gassett, Wiesler & Baker (1997)* | GC-MS | forehead, back | only volatiles analyzed; composition varied among individuals (e.g., different concentrations of decane, carvone, & undecane) & between forehead & back (e.g., different concentrations of terpene, octanal, & naphthalene) |

**Table 1** (*continued*)

| Reference | Method | Body Region | Conclusion |
|---|---|---|---|
| Colton et al. (1986) | TLC | mid-section | wax monoesters are the most common non-polar lipid on mink skin |
| Waldorf & Vedros (1978) | GC- FAME | Unknown | only fatty acid components identified; some were fungiostatic to dermatophytes *in vitro* |
| Williams et al. (1992) | LC- MS, TLC, GC- MS | Dorsal body, head, tail | lipid quantity varies among body parts (e.g., greater abundance on lower back versus head). Squalene is the principal lipid component of both fur & skin with minor amounts of other compounds such as triglycerides & cholesterol. |
| Davis et al. (1988) | GC- MS | Thorax, abdomen | only squalene examined; mean 3.7 1.1 mg/g of fur. Exposure to crude oil did not change squalene concentration, but cleaning fur with Dawn soap removed it completely. Concentrations returned to base-line levels 7 days after cleaning in unoiled but not oiled fur |
| Lindholm & Downing (1980) | TLC | Unknown | lipid quantity: 14mg/g of fur; found squalene, glyceryl ether diesters, unidentified polar lipid, free sterol, wax esters, & diesters |
| | | | lipid quantity: 3mg/g of fur; found squalene, unidentified polar lipid, free sterol, wax esters, & diesters |
| | | | lipid quantity: 30mg/g of fur; found squalene, unidentified polar lipid, free sterol, wax esters, & diesters |

**Notes.**

GC, gas chromatography; TLC, thin-layer chromatography; FAME, fatty acid methyl ester; FFA, free fatty acids; MS, mass spectrometry; LC, liquid chromatography; MALDI-TOF, Matrix-assisted laser desorption/ionization-time of flight.

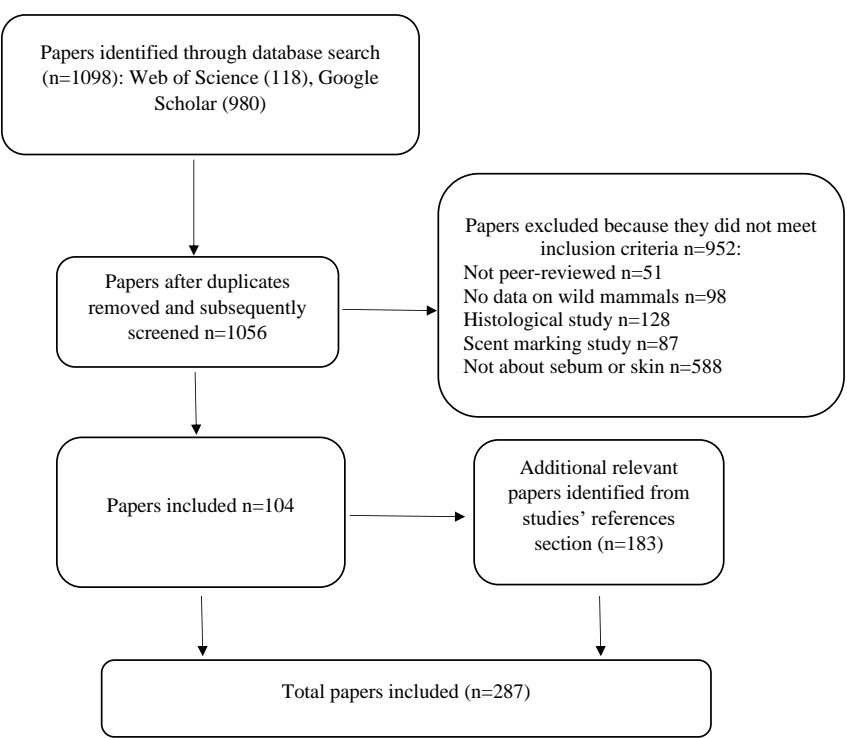

**Figure 2** **Protocol for screening articles after database search.** Each step shows the number of papers included or excluded for review.

# SEBACEOUS GLAND OCCURRENCE AND ANATOMY

## Anatomy

Sebaceous glands are composed of sebum-producing cells (sebocytes) that release their contents onto the skin surface *via* hair canals (*Thody & Shuster, 1989*; *Zouboulis et al., 2008*). Sebocytes undergo a maturation process followed by a cell-type specific death that results in the holocrine secretion of sebum (*Thody & Shuster, 1989*; *Zouboulis et al., 2008*). Sebaceous glands are usually found in association with hair follicles forming a pilosebaceous unit, with the sebaceous gland located in the upper portion of the hair follicle where it is not affected by the hair cycle (the four stages of growth and loss of hair) (*Smith & Thiboutot, 2008*). Some sebaceous glands occur without an associated hair follicle, such as the meibomian glands (eyelids) and Fordyce's spots (oral epithelium) (*Smith & Thiboutot, 2008*; *Zouboulis et al., 2016*). Sebaceous gland volume is partially determined by the surface area of the hair follicle, although not for those associated with vibrissae (*Haffner, 1998*). The amount of sebum produced at a particular time is governed by gland size and the number of secreting cells (*Sokolov, 1982*; *Makrantonaki, Ganceviciene & Zouboulis, 2011*). Ordinary sebaceous glands produce a continuous flow of sebum, resulting in constant lubrication of hair and skin (*Sokolov, 1982*). Changes in the composition of skin surface lipids have been used as an index of sebaceous gland activity. The palmitate-to-stearate and stearate-to-oleate ratios are positively correlated with sebaceous gland secretion rate in rats, and squalene synthesis rates may be positively correlated with gland size in humans (*Thody & Shuster, 1989*; *Nikkari & Valavaara, 1970*; *Strauss, Pochi & Downing, 1976*).

## Skin lipid composition

In humans and laboratory mammals sebum is generally composed of cell debris and nonpolar (neutral) lipids, namely triacylglycerol, diacylglycerol, wax esters, squalene, cholesterol, sterol esters, and free fatty acids (*Smith & Thiboutot, 2008*). These lipids also occur in other tissues or cell types (*Pappas, 2009*; *Smith & Thiboutot, 2008*). Lipids are generated not just by sebaceous glands, but also within the epidermis by keratinocytes (*Shi et al., 2015*). The composition of lipids produced by sebaceous glands and the epidermis differs despite some overlap (*Stewart & Downing, 1991*; *Pappas, 2009*; *Butovich, 2017*). Studies on the skin lipids of mammals do not always differentiate between epidermal and sebaceous lipids (*Pappas, 2009*) that complicates interpretation of lipid composition and function from different sources. Some studies included in this review may comprise a combination of epidermal lipids and sebum as research on general skin surface lipids. However, studies focused exclusively on epidermal lipids are not included in this review.

## Sebaceous gland occurrence

Sebaceous glands are absent in a number of species that are hairless or have a sparse distribution of fur or hair, including the Cetacea (whales, dolphins, porpoises), Hippopotamidae (hippos), Elephantidae (elephants), naked mole-rat (*Heterocephalus glaber*), and sirenians (Dugongidae, Trichechidae) (*Daly & Buffenstein, 1998*; *Springer & Gatesy, 2018*; *Lopes-Marques et al., 2019*; *Menon et al., 2019*; *Springer et al., 2021*). In rhinoceros species sebaceous glands are either absent or poorly developed (*Springer &*
*Gatesy, 2018*). In pangolins (*Manis* spp.) and desert hedgehogs (*Paraechinus aethiopicus*) sebaceous glands are restricted to the snout and abdomen (*Springer & Gatesy, 2018*; *Massoud, 2020*). Sebaceous glands are also absent in some species with fur, namely Cynocephalidae (colugos) (*Springer & Gatesy, 2018*), but further information on colugo's skin properties is unavailable. Aside from these exceptions, sebaceous glands are nearly ubiquitous, though unevenly distributed, in the hair-bearing skin of most mammals.

## Scent glands

Many mammalian species have scent glands composed of enlarged and modified sebaceous glands that produce chemical signals communicating information about species, sex, individual identity, reproductive condition, and social status (*Zouboulis et al., 2008*). Much of the literature on sebaceous glands in wild mammals focuses on scent glands and patterns in these studies may provide insight into non-specialized sebaceous glands. The composition of secretions from scent glands varies with reproductive status, social status, body condition, season, sex, diet, and age in a variety of wild mammals, and there can be overlap with the lipid composition from sebaceous glands, but often the composition is different (*Adams, Li & Wilkinson, 2018*; *Buesching, Newman & Macdonald, 2002*; *Buesching, Waterhouse & Macdonald, 2002*; *Burger et al., 1999*; *Dingzhen et al., 2006*; *Faulkes et al., 2019*; *Gassett et al., 1996*; *Jenkinson, Blackburn & Proudfoot, 1967*; *Kannan & Archunan, 1999*; *Khazanehdari, Buglass & Waterhouse, 1996*; *Martín et al., 2014*; *Muñoz Romo et al., 2012*; *Nassar et al., 2008*; *Nikkari, 1974*; *Rasmussen, 1988*; *Rossini & Ungerfeld, 2016*; *Salamon, Davies & Stoddart, 1999*; *Salamon & Davies, 1998*; *Sergiel et al., 2017*; *Sokolov et al., 1980*; *Thody & Shuster, 1989*; *Volkman, Zemanek & Muller-Schwarze, 1978*; *Waterhouse et al., 1996*; *Wood et al., 2005a*; *Wood et al., 2005b*; *Zabaras, Wyllie & Richardson, 2005*). Scent glands are often sexually dimorphic, reflecting differences in breeding activity and responses to sex steroid hormones, but scent gland secretion composition does not vary by sex in all species (*Woolhouse, Weston & Hamilton, 1994*; *Forman, 2005*; *Burger et al., 2020*). Scent glands may also play a role in pathogen defense, thermoregulatory responses, and maintaining skin barrier function (*Quay, 1970*; *Forman, 2005*).

Research on sebum function has focused on the maintenance of healthy skin and defense against pathogens, while research on scent gland function has focused on chemosensory communication. The functions of these two gland types may overlap more than is currently recognized. Non-specialized sebaceous glands may play a role in chemosensory communication *via* delivery of pheromones to the skin surface, although the contribution of these glands to communication relative to specialized scent glands is unclear (*Smith & Thiboutot, 2008*). A chemosensory function may help explain the specificity of each species' sebum composition. Scent glands may help maintain skin health as components of scent gland secretions may have insecticidal properties that could reduce ectoparasite loads (*Muñoz Romo et al., 2012*). Further studies on the function of scent glands in mammals beyond those for communication are needed, as they may play an important role in disease occurrence and progression.

# FUNCTIONS OF SEBUM

Changes in sebum composition or quantity can be a cause or consequence of disease and impact the functions sebum performs. For instance, compromised skin barrier function, which is assessed by measuring rates of cutaneous water loss, is an indicator of various skin disorders (*Ohman & Vahlquist, 1994*; *Muñoz Garcia et al., 2012*; *Knox & O'Boyle, 2021*). Understanding how sebum functions in healthy conditions may provide insight into disease mechanisms. Below we review functions performed by sebum that are important in maintaining healthy skin.

## Skin barrier function

A major challenge of terrestrial wildlife is to minimize water loss, especially in dry environments. If mammalian skin is damaged or diseased, cutaneous water loss can increase by several orders of magnitude (*Lillywhite, 2006*). Dry skin is linked to various forms of dermatitis and in extreme cases excessive cutaneous water loss can lead to death from dehydration (*Nishifuji & Yoon, 2013*). Dry skin can crack and disrupt the skin barrier, which then provides entry points for microbes (*Nishifuji & Yoon, 2013*). Epidermal lipids, rather than sebaceous lipids, are thought to play the dominant role in minimizing cutaneous water loss (*Lillywhite, 2006*). Compared to sebaceous lipids, epidermal lipid composition is similar among mammalian species, possibly due to evolutionary conservation of a fundamental mechanism for water retention, although data on only a few species are available (*Nicolaides, Fu & Rice, 1968*; *Birkby, Wertz & Downing, 1982*; *Wertz, Colton & Downing, 1983*). Sebum can also contribute to waterproofing the skin. Among armadillo species the greater development of sebaceous glands in Euphractinae compared to Dasypodinae is thought to prevent desiccation of cornified scales in extremely arid climates (*Krmpotic et al., 2015*). Laboratory mice genetically engineered to have defective or missing sebaceous glands have disrupted hair cycles, dry hair, skin lesions, defective water repulsion, compromised thermoregulation, and chronic, progressive alopecia (hair loss) (*Wood et al., 2005a*; *Wood et al., 2005b*; *Zhang et al., 2014*). Laboratory mice with intact sebaceous glands mutated to lack various enzymes and proteins important for lipid metabolism and secretion on the skin surface developed atrophic sebaceous glands, defective production of skin lipids, and altered lipid composition accompanied by dry/brittle fur and hair loss (*Chen et al., 2002*; *Zhang et al., 2014*; *Westerberg et al., 2004*). After water immersion these mice also exhibited impaired water repulsion, increased rate of trans-epidermal water loss, and hypothermia (*Chen et al., 2002*; *Zhang et al., 2014*; *Westerberg et al., 2004*). Sebaceous gland degeneration is characteristic of some types of alopecia in humans and laboratory mice (*Schneider & Zouboulis, 2018*; *Smith & Thiboutot, 2008*; *Pappas, 2009*). *Asebia* mutated mice are characterized by sebaceous gland hypoplasia induced through spontaneous mutation of the gene *ab* (*Schneider, 2015*). This mutation impairs production of glycerol, a contributor of stratum corneum hydration, which emphasizes the importance of glycerol generation from triglycerides in the sebaceous glands (*Fluhr et al., 2003*). Lipids from both sebaceous glands and the epidermis likely contribute to minimizing water loss, particularly since sebum can impact epidermal lipid metabolism and expression (*Ludovici*
*et al., 2018*). These findings illustrate the importance of sebum in maintaining healthy skin and hair as part of the host's defenses against disease.

Sebum also plays a role in thermoregulation, where hypothermia or hyperthermia can result in death (*Cheshire, 2016*). At higher temperatures, sebum acts as a surfactant for eccrine secretions in humans to retain sweat and promote heat loss, as sweat that immediately drips off the skin does not effectively dissipate heat (*Nicolaides, Fu & Rice, 1968*; *Porter, 2001*). At lower temperatures, in its viscous form, sebum acts as a local repellent of rain on exposed skin (*Butcher & Coonin, 1949*). Therefore, the outcome of secretory interactions is for an external fluid, rain, to be projected off the skin in cool wet conditions, whereas in hot conditions, the internally generated fluid, eccrine sweat, is encouraged to spread in a film across the skin and be retained on the surface (*Butcher & Coonin, 1949*; *Nicolaides, Fu & Rice, 1968*; *Porter, 2001*). It is unclear if this is an important thermoregulatory mechanism in hot conditions in other mammals, as sweating is best known in humans (eccrine) and horses (apocrine), despite also occurring in a diminished capacity in other mammals (*Robertshaw, 1985*). Similar to humans, sebum creates a water-repellent pelage (hair, fur, or wool) in wild mammals by coating hair and fur to prevent over-wetting and resulting hypothermia (*Porter, 2001*; *Thody & Shuster, 1989*; *Walro & Svendsen, 1982*; *Waldorf & Vedros, 1978*; *Zhang, Chaturvedi & Chaturvedi, 2015*). Effectively repelling water off the skin is not just important for thermoregulation. Excessive wetting softens the skin and disrupts normal cutaneous microflora, which can increase disease susceptibility (*Tellam et al., 2021*).

While sebum-coated fur is important to repel water, excess lipids can cause fur to mat, thus compromising insulative qualities (*Harriman & Thiessen, 1983*). Some species of rodents, such as kangaroo rats (*Dipodomys* spp.) and gerbils (*Meriones* spp.), groom and sandbathe to remove excess lipids from their fur (*Randall, 1981*; *Thiessen & Pendergrass, 1985*). Captive Mongolian gerbils (*Meriones unguiculatus*) living at 10 °C had significantly higher levels of pelage lipids than at 24 °C, suggesting a role of lipids in thermoregulation (*Thiessen & Pendergrass, 1985*). Individuals can alter pelage lipid quantity by either removing lipids through sandbathing or increasing lipids by autogrooming secretions from Harderian glands (*Thiessen & Pendergrass, 1985*). Harderian glands are present in a variety of mammals and are located near the eyes (*Sakai, 1981*). Removing Harderian glands, or shampooing animals, decreases the quantity of pelage lipids and decreases the ability of individuals to thermoregulate in cold environments, while increasing evaporative water loss in hot environments in both gerbils and muskrats (*Ondotra zibethicus*) (*Thiessen & Pendergrass, 1985*; *Thiessen & Kittrell, 1980*; *Harlow, 1984*). Thermoregulation is restored and evaporative water loss decreased by applying lipids or mineral oil to the skin (*Thiessen & Kittrell, 1980*; *Harlow, 1984*). This mechanism may also exist in other mammals, such as blind mole rats (*Nannospalax ehrenbergi*) (*Shanas & Terkel, 1996*).

Except for marsupials (*Ferner, 2021*), and species lacking sebaceous glands, fetal sebaceous glands activate during gestation and in humans they produce vernix caseosa, a white lipid-rich biofilm covering the skin, in the last trimester of pregnancy (*Shannon, 2020*). Vernix contains both sebaceous lipids and epidermal lipids produced by the fetus (*Hoath, Pickens & Visscher, 2006*; *Nishijima et al., 2019*). The biological function of

vernix caseosa is not well understood but is thought to be a barrier to water loss, assist thermoregulation after birth, have antimicrobial and anti-oxidant functions, facilitate skin surface acidification, and potentially act as a film to minimize friction during delivery (*Visscher et al., 2005*; *Hoath, Pickens & Visscher, 2006*; *Wang et al., 2018*; *Nishijima et al., 2019*; *Shannon, 2020*). The vernix lipid composition of California sea lions (*Zalophus californianus*), the only other mammal aside from humans known to produce vernix, is similar to human vernix (*Wang et al., 2018*).

## Protection against abiotic stressors

A major challenge for terrestrial wildlife is protecting skin against oxidative stressors such as ultraviolet radiation, ozone, and chemicals. Oxidative stress regulates major signaling pathways of extrinsic skin aging and skin diseases like acne, various forms of dermatitis, and skin carcinogenesis (*Briganti & Picardo, 2003*; *Masaki, 2010*; *Zouboulis et al., 2016*). Skin that is damaged by these stressors is more prone to infection because barrier function can be compromised (*Zouboulis et al., 2016*). Sebum provides photoprotection, but exposure to UV radiation can lead to cellular damage by changing the composition of skin lipids, such as increasing the percentage of free fatty acids and cholesterol in humans and laboratory rodents (*Gloor & Karenfeld, 1977*; *Ohsawa et al., 1984*; *Picardo et al., 1991*; *Marques et al., 2002*; *Akitomo et al., 2003*; *Mudiyanselage et al., 2003*; *Zouboulis et al., 2016*). Exposure to UV radiation can also increase the amount of skin surface lipids from both epidermal lipids and sebaceous glands depending on the dosage (*Gloor & Karenfeld, 1977*; *Akitomo et al., 2003*). Exposure to environmental pollutants and toxins can change skin lipid composition and inhibit lipogenesis in human sebaceous glands, and sebum is one of the skin's defenses against such toxins (*Zouboulis et al., 2016*). Human sebaceous glands secrete vitamin E onto the upper layers of the skin which is protective again oxidation (*Thiele, Weber & Packer, 1999*), but this has not been studied in wildlife.

## Microbes and sebum

Mammalian skin lipid composition can affect microbial growth, attachment to skin, and the production of virulence factors, but skin lipids can also be an important nutrient source for both commensal and pathogenic microbes (*Drake et al., 2008*; *Fischer et al., 2014*). Diverse microhabitats across skin surfaces affects the density and diversity of microbial colonization, including pathogens (*Kearney et al., 1984*; *Harder, Schröder & Gläser, 2013*). Variation in skin microhabitats are caused by morphological differences, such as presence of hair and glands, which cause variations in temperature, pH, moisture, nutrient availability, and the composition of antimicrobial peptides and lipids (*Kearney et al., 1984*; *Grice et al., 2009*; *Findley et al., 2013*; *Schommer & Gallo, 2013*). Microbes attempting to colonize skin surfaces must attain nutrients from either lipids, skin cells, other microbes, or hair on the skin surface, as well as contend with skin antimicrobial properties (*Mukherjee et al., 2016*; *Roux, Oddos & Stamatas, 2021*). Substantial microbial populations occur in sebaceous glands and associated hair follicles in humans and domestic mammals (*Harder, Schröder & Gläser, 2013*; *Kearney et al., 1984*; *Naik et al., 2012*). The prevalence and composition of microbes on sebaceous-rich skin sites in

humans, such as the face and upper body, differs from dry sites such as the forearm and buttock (*Sanmiguel & Grice, 2015*; *Mukherjee et al., 2016*). In humans, the stimulation of sebaceous gland secretion by hormones at puberty favors lipophilic taxa on the skin, such as *Corynebacterium* spp. and *Cutibacterium* spp., that are considered normal components of skin microbiomes (*Mukherjee et al., 2016*; *Roux, Oddos & Stamatas, 2021*). The skin microbiome plays a role in host defense against pathogens (*Chen, Fischbach & Belkaid, 2018*). While sebum quantity and composition influences the skin microbiome (*Pyle et al., 2023*), microbes can also alter sebum composition. Bacteria secrete lipases which break down triglycerides secreted from sebaceous glands (*Zouboulis, 2004*; *Drake et al., 2008*). Bacteria can also alter the composition of scent gland secretions in multiple wild mammal species through fermentation and breaking down proteins and carbohydrates (*Osborn et al., 2000*; *Woolhouse, Weston & Hamilton, 1994*; *Studier & Lavoie, 1984*; *Albone et al., 1974*; *Burger et al., 1999*; *Voigt, Caspers & Speck, 2005*; *Theis et al., 2013*; *Gonzalez-Quinonez, Fermin & Munoz-Romo, 2014*). Impaired production or alteration of sebum composition have been proposed as key features in atopic dermatitis and susceptibility to microbial colonization (*Zouboulis, 2004*; *Fischer et al., 2014*; *Knox & O'Boyle, 2021*). Components of human and laboratory mouse sebum, particularly fatty acids like lauric acid, oleic acid, sapienic acid, and palmitoleic acid, reduce growth of various pathogenic gram-positive bacteria, block adhesion to skin by fungi, and prevent germination of various dermatophytes (*Bibel, Aly & Shinefield, 1992*; *Wille & Kydonieus, 2003*; *Georgel et al., 2005*; *Drake et al., 2008*; *Chen et al., 2011*; *Fischer et al., 2014*). For instance, skin deficient in free fatty acids is more susceptible to colonization by the opportunistic pathogen, *Staphylococcus aureus*, and protection against colonization is bolstered with the application of topical fatty acids (*Georgel et al., 2005*; *Takigawa et al., 2005*). Free fatty acids are produced *via* hydrolysis of their precursors, triglycerides secreted from sebaceous glands, by lipases secreted from commensal bacteria such as *C. acnes* and *Staphylococcus epidermidis* and by acid lipase produced by the epidermis (*Zouboulis, 2004*; *Drake et al., 2008*). These findings illustrate that skin lipid composition influences skin microbiome composition and function, as well as disease susceptibility.

Free fatty acids may provide direct antimicrobial activities against bacteria and enhance the skin's innate antimicrobial defense by inducing the expression of human $\beta$-defensin-2, an antimicrobial peptide, in human sebocytes and mouse skin (*Nakatsuji et al., 2010*). Antimicrobial peptides and lipids on the skin can act synergistically against bacteria and yeast (*Robertson et al., 2006*; *Fischer et al., 2014*). Free fatty acids inhibit bacterial growth or induce death by cell lysis, inhibition of enzyme activity, impairment of nutrient uptake, and the generation of toxic peroxidation and autooxidation products (*Desbois & Smith, 2010*). However, some skin pathogens, such as *Staphylococcus aureus*, are able to detoxify specific skin antimicrobial fatty acids (*Subramanian et al., 2019*). Besides inhibiting or killing bacteria directly, free fatty acids also make conditions unfavorable for the growth of certain bacteria on the skin surface by maintaining an acidic pH (*Fluhr et al., 2001*; *Takigawa et al., 2005*). The antimicrobial activity of skin lipids varies with pH *in vitro*, with almost no activity >8pH (*Bibel, Aly & Shinefield, 1992*). Sebaceous glands can synthesize and secrete either pro- and anti-inflammatory cytokines and lipids in response to environmental

stimuli, such as the presence of microbes (*Zouboulis, 2004*; *Zouboulis et al., 2008*; *Lovászi et al., 2018*).

Most information available on the antimicrobial properties of wild mammal sebum derives from bats in North America. Recent research on the skin lipids of bats was prompted by the discovery of the fungus *Pseudogymnoascus destructans* (*Pd*) that causes white-nose syndrome (*Lorch et al., 2011*). The *Pd* hyphae can penetrate both the epidermis and dermis, causing severe skin lesions and destroying hair follicles, sebaceous glands, and sweat glands (*Meteyer et al., 2009*; *Meteyer et al., 2022*; *Courtin et al., 2010*). Research on the role of sebum in white-nose syndrome has focused on the antimicrobial properties of sebum against *Pd in vitro* rather than the disease itself. Sebum composition varies among bat species (*Frank et al., 2016*; *Pannkuk et al., 2012*) and changes in both composition and quantity over the hibernation season, both of which may affect *Pd* growth (*Frank et al., 2016*; *Frank et al., 2018*; *Ingala et al., 2017*). Infection with *Pd* changes the lipid composition of wing tissue (*Pannkuk et al., 2015*). Some skin lipids of little brown myotis bats (*Myotis lucifugus*) and big brown bats (*Eptesicus fuscus*), such as 1-monopalmitolein, behenyl palmitoleate (wax ester), palmitoleic acid, pentadecanoic acid, linoleic acid, and stearic acid, inhibit *Pd* growth *in vitro* (*Ingala et al., 2017*; *Frank et al., 2018*), but *Pd* growth and inhibition results differ depending on incubation temperature and media composition (*Frank et al., 2016*; *Ingala et al., 2017*; *Gabriel et al., 2019*). The ability of some bats species, such as *E. fuscus*, to resist or tolerate *Pd* infection may be partially due to the wax ester, free fatty acid, and 1-monoacylglycerol composition of their skin lipids (*Frank et al., 2016*; *Frank et al., 2018*). The epidermis of *E. fuscus* contains almost twice as much myristic, palmitoleic, and oleic acids as *M. lucifugus*, a white-nose syndrome-susceptible bat species, and these compounds all inhibit *Pd* growth *in vitro* (*Frank et al., 2016*). Sebum from *M. myotis*, a European bat species that is highly resistant to cutaneous *Pd* infections, contains over 120 distinct types of wax esters (*Řezanka et al., 2015*), some of which inhibit *Pd* growth *in vitro* (*Frank et al., 2018*). Although *Pd* is not lipophilic, the fungus releases lipases, esterases, and proteinases (*Raudabaugh & Miller, 2013*; *Reynolds & Barton, 2014*). Hyphae are consistently seen at the openings of hair follicles and within sebaceous glands in infected bats (*Meteyer et al., 2022*). Aside from bats, information on the antimicrobial properties of wild mammal sebum is available for only two other species. The free fatty acid portion of lipids that coat porcupine (*Erethizon dorsatum*) quills inhibits some bacteria strains *in vitro* (*Roze, Locke & Vatakis, 1990*). Some fatty acids from northern fur seal (*Callorhinus ursinus*) skin, such as oleic acid and stearic acid, inhibited growth of five dermatophyte species *in vitro* (*Waldorf & Vedros, 1978*). As illustrated by white-nose syndrome, the antimicrobial properties of sebum are likely important in multiple skin diseases of wild mammals.

## How essential is sebum?

The importance of sebum for skin health in humans has been questioned because the sebaceous glands of prepubescent children are largely inactive and because the skin on adults' palms and soles lacks sebaceous activity but functions well (*Kligman, 1963*; *Stewart & Downing, 1991*). Sebum production in humans is high at birth, which can lead to acne, but sebaceous glands shrink during childhood until puberty (*Shannon, 2020*). Multiple

forms of dermatitis disappear with the onset of puberty and accompanying increase in sebaceous gland activity (*Rothman, Smijanic & Weitkamp, 1946*; *Shi et al., 2015*; *Wertz, 2018*). It is unknown how active sebaceous glands must be to minimize water loss, support thermoregulation, protect against pathogens, and prevent UV-induced damage. Sufficient sebum may be produced by children to fulfill these functions (*Stewart & Downing, 1991*).

The lack of sebaceous glands in some mammalian lineages also implies that sebum may not be essential to skin function (*Daly & Buffenstein, 1998*; *Springer & Gatesy, 2018*; *Lopes-Marques et al., 2019*; *Menon et al., 2019*; *Springer et al., 2021*). Species lacking sebum are characterized by a sparse distribution or absence of hair and fur except for colugos (order Dermoptera). Skin oils are absorbed by fur in most mammals. However, in hairless species with sebaceous glands, oils remain on the skin and can cause problems. For example, humans and hairless (Sphynx) cats have normal sebaceous glands and sparse or thin hair, and consequently can have oily or greasy skin with associated skin problems such as acne (*Ahman & Bergström, 2009*; *Genovese et al., 2014*). Hairless species such as rhinos and naked mole rats that lack sebaceous glands may benefit by avoiding such problems, but is unclear how they replace the beneficial functions performed by sebaceous glands in other mammals. Potential strategies include regular wetting or immersion of the skin in water, to prevent dry skin, secretions from other glands with UV-protection and antimicrobial properties, and increased rates of epidermal desquamation to prevent colonization by microorganisms, ectoparasites, and macrosymbionts (*Eltringham, 1999*; *Saikawa et al., 2004*; *Martinez-Levasseur et al., 2011*; *Martinez-Levasseur et al., 2013*; *Lillywhite & Stein, 1987*; *Brown et al., 1983*; *Hicks et al., 1985*; *Fish & Hui, 1991*).

## NON-HUMAN, MAMMALIAN SKIN DISEASES ASSOCIATED WITH SEBACEOUS GLANDS

Below we review all known skin diseases associated with sebum and sebaceous glands in wild mammals, apart from white-nose syndrome as discussed above. We highlight potential functions of sebum in the prevention or exacerbation of disease, and various consequences that can occur when normal functions performed by sebum are disrupted (Fig. 3).

### Dermatophilus congolensis and Dermatophilosis

Dermatophilosis affects a wide range of domestic and wild mammalian species, including ungulates, rodents, bears, mustelids, monkeys, primates, and pinnipeds, although most knowledge of the disease derives from research on domestic sheep (*Montali et al., 1981*; *Salkin & Gordon, 1983*; *Zaria, 1993*; *Brack et al., 1997*; *Nemeth et al., 2014*; *Ayalew et al., 2015*; *Caron et al., 2018*). It can cause major economic losses to livestock owners, owing to the downgrading of skin/wool, lower meat and milk production, and mortality of stock (*Zaria, 1993*; *Msami et al., 2001*; *Ayalew et al., 2015*). Case fatality rates for dermatophilosis vary from 10–50% in some domestic species (*Gitao, Agab & Khalifalla, 1998*; *Ayalew et al., 2015*). Mortality rates and non-lethal effects have not been quantified in wild mammals (*Zaria, 1993*). The effect of dermatophilosis on wild mammal populations may resemble domestic mammals, or it may differ due to lower host densities or other skin properties.

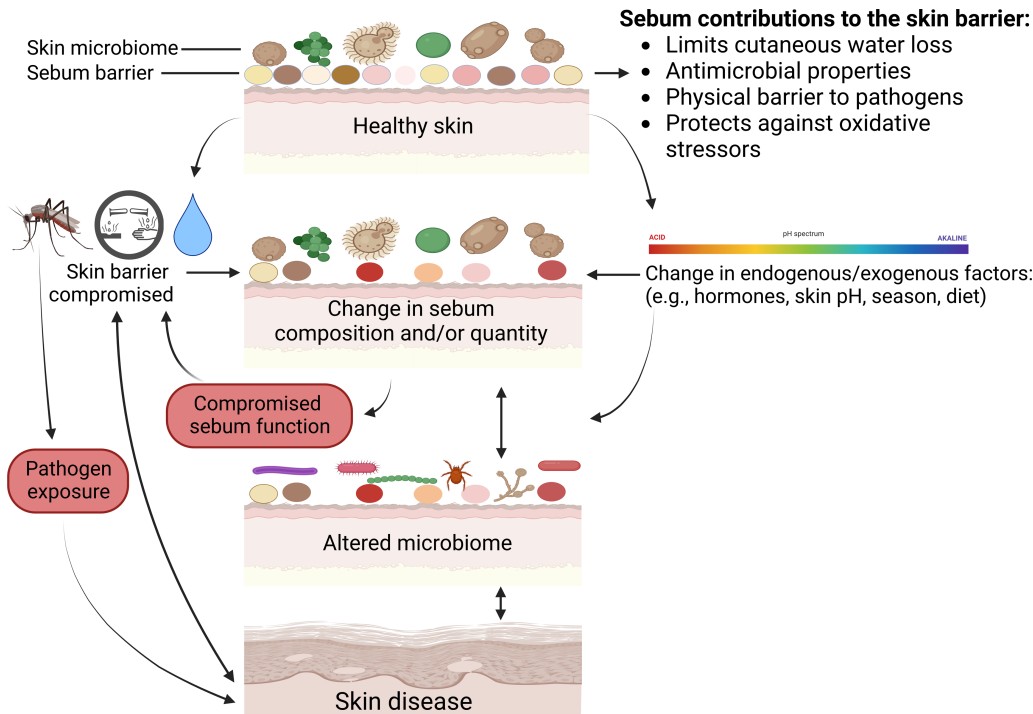

**Figure 3** **Mechanisms on how skin defenses with regards to sebum can fail and result in skin disease.** Physical injuries such as cuts or insect bites can bypass skin defenses, and chemical insults or excessive exposure to water can compromise the sebum barrier or change skin microbiomes. Microbiomes comprise bacteria, fungi, and viruses. Multiple factors can change sebum composition or quantity that in turn can alter skin microbiomes or sebum function and lead to disease. Skin diseases can further modify skin microbiomes, change sebum, and compromise skin barrier function. Created with BioRender.com.

The actinomycete bacterium *Dermatophilus congolensis* causes the skin disease dermatophilosis that presents as skin lesions characterized by an exudative dermatitis (*Zaria, 1993*; *Ayalew et al., 2015*). *Dermatophilus congolensis* is not highly invasive and does not normally breach the barriers of healthy skin (*Zaria, 1993*; *Ayalew et al., 2015*). It is considered a normal component of cutaneous microflora and likely requires a compromised skin barrier, such as minor wounds or transmission *via* insect bites, as a precursor to active infection (*Zaria, 1993*). During infection, *D. congolensis* invades the keratinized layer of the skin along with hair follicles and sebaceous glands (*Roberts, 1967*). *Dermatophilus congolensis* secretes proteins, especially proteases to aid removal of the protective outer keratin layer of skin, lipases to remove skin lipids, and haemolysins to allow bacterial invasion of cells, that collectively facilitate invasion of the skin (*How, Lloyd & Sanders, 1990*). Infection rates are higher in young animals, potentially because skin barrier function is compromised since the skin lipid layer is not yet properly formed (*Roberts, 1963a*).

Increased rain and humidity leading to persistent wetting of the hair and skin are key environmental factors associated with *D. congolensis* infection (*Zaria, 1993*; *Tellam et al., 2021*). The disease has a worldwide distribution but is most prevalent in humid tropical

and subtropical regions, with mortality peaking during the rainy season (*Zaria, 1993*; *Ayalew et al., 2015*). Lesion distribution in some species is concentrated in body regions such as the back that are prone to direct rain exposure (*Le Riche, 1968*; *Dalis et al., 2009*). Prolonged exposure to moisture can disperse the protective lipid layer on the skin, change lipid composition, softens the skin, and disrupts normal cutaneous microflora, thereby increasing skin vulnerability to *D. congolensis* infection in sheep (*Tellam et al., 2021*; *Colditz et al., 2021*; *James, Warren & Neville, 1984*; *Hay & Mills, 1982*). Moisture also promotes *D. congolensis* infection by causing the release of infective zoospores from infected scabs (*Roberts, 1963b*).

The mechanical properties of the sebaceous film as a barrier to *D. congolensis* and water are potentially more important in resisting infection than sebum's bacteriostatic action (*Roberts, 1963a*). Experimental infection of domestic sheep with *D. congolensis* without removing the sebaceous film produces only scattered lesions (*Roberts, 1963a*). Studies that experimentally challenge skin with *D. congolensis* generally remove skin lipids before the addition of spores (*Roberts, 1963a*; *Le Riche, 1968*; *Tellam et al., 2021*). Aside from antibiotics and vaccines, a topical treatment, Lamstreptocide, for the disease consists of sebaceous fatty acids such as palmitic, stearic, oleic, and linoleic acid (*Zaria, 1993*; *Ayalew et al., 2015*). These results illustrate the protective properties of sebum against pathogens. *Dermatophilus congolensis* may also be inhibited by commensal microbes on the skin (*Kingali, Heron & Morrow, 1990*; *Zaria, 1993*).

## Malassezia spp. and Dermatitis

The genus *Malassezia* consists of 18 species of dimorphic lipophilic yeasts that are common components of the mammalian skin microbiome (*Guillot & Bond, 2020*; *Batra et al., 2005*). They are considered opportunistic skin pathogens, although causal relationships of *Malassezia* species with dermatological disorders are sometimes unclear (*Guillot & Bond, 2020*; *Batra et al., 2005*). The genus is associated with skin conditions in humans such as dandruff, seborrheic dermatitis, atopic dermatitis, *Malassezia* folliculitis, psoriasis, and pityriasis versicolor (*Gueho et al., 1998*; *Ashbee & Evans, 2002*; *DeAngelis et al., 2005*; *Harada et al., 2015*; *Theelen et al., 2018*). Skin conditions associated with *Malassezia* often improve with anti-fungal treatment, which supports causal relationships of *Malassezia* with these skin disorders (*Plant, Rosenkrantz & Griffin, 1992*; *Harada et al., 2015*). *Malassezia* dermatitis and otitis is common in dogs but is also found in other domesticated animals such as cats, pigs, cattle, horses, and goats (*Guillot & Bond, 2020*; *Batra et al., 2005*). Hairless (Sphynx) cats are known for oily/greasy skin and have higher rates of *Malassezia* carriage compared to other cat breeds (*Ahman & Bergström, 2009*).

Research on *Malassezia* in wildlife has documented the genus on the skin of various wild mammals but has not explored its association with skin disorders. It has been isolated from free-ranging species with sarcoptic mange such as red fox (*Vulpes fulva*), porcupine (*Erethizon dorsatum*), and coyote (*Canis latrans*), and also from zoo animals with dermatitis such as Indian rhinoceros (*Rhinoceros unicornis*), white rhinoceros(*Ceratotherium simum simum*), South American sea lions(*Otaria byronia*), gray seal (*Halichoerus grypus*), harbor seal (*Phoca vitulina*), and California sea lions (*Zalophus californianus*) (*Guillot et al., 1998*;

*Pollock, Rohrbach & Ramsay, 2000*; *Hadjina et al., 2019*; *Salkin, Stone & Gordon, 1980*; *Nimmervoll et al., 2013*; *Weidman, 1925*; *Bauwens, De Vroey & Meurichy, 1996*; *Nakagaki et al., 2000*). However, *Malassezia* spp. are also present on a variety of free-ranging and captive mammal species with no skin disease (*Lorch et al., 2018*; *Gandra et al., 2008*; *Coutinho et al., 2020*; *Neves et al., 2017*; *Wesche & Bond, 2003*; *Kuttin & Müller, 1994*; *Dall' Acqua Coutinho, Fedullo & Corrêa, 2006*). Given its occurrence in domestic mammals, *Malassezia* dermatitis and otitis likely also occur in wild mammals. Therefore, we summarize the interplay of *Malassezia* and skin lipids on human and domestic mammal skin below, as mechanisms may be similar in wild mammals.

*Malassezia* species cannot produce fatty acids themselves and require lipids from the environment for growth (*Theelen et al., 2018*). *Malassezia* releases lipases, phospholipases, aspartyl proteases, and acid sphingomyelinases that hydrolyze lipid sources like sebum triglycerides to obtain fatty acids (*Ashbee & Evans, 2002*; *Celis et al., 2017*). These enzymes enable growth of these yeasts on host skin and change host sebum composition (*Celis et al., 2017*). The unsaturated free fatty acids hydrolyzed from triglycerides by *Malassezia*, such as oleic acid and arachidonic acid, can result in inflammation, irritation, scaling, and skin barrier defects in susceptible human individuals (*Ashbee & Evans, 2002*; *Gupta et al., 2004*; *DeAngelis et al., 2005*; *Ro & Dawson, 2005*; *Harada et al., 2015*). Indeed, applying oleic acid to human scalps can induce flaking in dandruff-susceptible but not non-susceptible individuals (*DeAngelis et al., 2005*). *Malassezia* interact with their host directly *via* chemical mediators and indirectly through immune interplay, so both host immunity and host barrier function have roles in *Malassezia*-associated skin disorders (*Wikramanayake et al., 2019*).

Since sebum is an important nutrient source for *Malassezia*, diseases that cause increased sebum production, such as some endocrine and bacterial skin diseases, provide a cutaneous microenvironment that encourages overgrowth of *Malassezia* spp. (*Batra et al., 2005*). Although *Malassezia* can be found on skin with limited sebum, such as the toe-web space and palms in humans, it is most abundant on body parts rich in sebum. Sebum-rich parts in humans include the face and scalp that are also the most common areas for skin disorders associated with *Malassezia*, such as seborrheic dermatitis and pityriasis versicolor (*Gueho et al., 1998*; *Findley et al., 2013*; *Harada et al., 2015*; *Jo, Kennedy & Kong, 2017*). In humans, age and sex are associated with changes in *Malassezia* composition on the skin as well as *Malassezia*-associated skin disorders, likely due to differences in the activity of sebaceous glands driven by hormones (*Ashbee & Evans, 2002*; *Ro & Dawson, 2005*). Other disturbances of skin microenvironmental factors, such as temperature, humidity, and skin pH, can also contribute to the development of dermatomycosis (*Hadjina et al., 2019*).

## Cutibacterium acnes and acne

Acne is a good case study for illustrating the role of sebum in skin diseases because it is a well-known skin disorder closely associated with sebum production and composition. Further work with more diverse species is needed to confirm that humans are a reasonable study system for the role of sebum in mammals. Acne is primarily a human disease, although minor forms of acne occur in dogs and cats (*Shannon, 2020*). This may be partially due

to the differences in sebum composition among species. For example, sapienic acid is a sebaceous fatty acid unique to humans and is implicated in the development of acne (*Shannon, 2020*). Sebum composition on skin with acne differs from unaffected skin, as patients produce sebum with more squalene and decreased levels of linoleic acid (*Pappas et al., 2009*; *Melnik, 2015*; *Shi et al., 2015*; *Knox & O'Boyle, 2021*). The pathogenesis of acne includes increased production of sebum (as occurs during adolescence in humans), blockage of the pilosebaceous unit, increased inflammation, and increased quantity of bacteria (*Zouboulis, 2004*; *Shi et al., 2015*; *Suh & Kwon, 2015*). Acne in dogs and cats primarily occurs on the chin, but the pathogenesis is largely unknown (*Plewig & Kligman, 2000*). Mexican hairless dogs can develop acne on multiple body parts, especially the limbs and back (*Kimura & Doi, 1996*).

The bacterium *Cutibacterium acnes* is associated with acne and is more prevalent on sebaceous body parts where sebum is its nutrient source (*Smith & Thiboutot, 2008*; *Shi et al., 2015*). Lipases and peroxidases produced by the bacteria cleave sebaceous triglycerides into glycerol and free fatty acids, such as palmitic acid, which are inflammatory, as well as oxidizing squalene (*Melnik, 2015*). Increases in palmitic and oleic acid on the skin are thought to drive comedogenesis and further microbial colonization of the skin (*Melnik, 2015*; *Lovászi et al., 2018*). Sebum composition affects *C. acnes* adherence and growth on the skin (*Melnik, 2015*). *Cutibacterium acnes* is not common on domestic animals, possibly due to sebum composition, but has been found on guinea pigs, cats, and dogs (*Webster, Ruggieri & McGinley, 1981*). The only report from wild mammals we are aware of is *Cutibacterium* sp. on a beaver (*Castor canadensis*) (*Rogovskyy et al., 2012*).

## Ectoparasites and demodicosis/dermatitis

A variety of ectoparasites, such as lice (*Trichodectes* spp.), feed on sebaceous secretions and can cause skin problems (*Jimenez et al., 2010*), but hair follicle mites (*Demodex* spp.), are specialized to live in sebaceous glands (*Izdebska, 2009*). *Demodex* spp. parasitize a wide range of domesticated and wild mammalian species (*Sastre et al., 2016*; *Jańczak et al., 2017*). Mites occupy the sebaceous gland portion of the pilosebaceous complex and feed on sebum and epithelia, generally without causing any clinical signs such as inflammation or lesions (*Mauldin & Peters-Kennedy, 2015*; *Jańczak et al., 2017*). The greatest concentration of mites occurs in areas of the body rich in sebaceous glands (*Jimenez-Acosta, Planas & Penneys, 1989*; *Mauldin & Peters-Kennedy, 2015*). Mites can become pathogenic when they proliferate excessively in response to changes in the host's cutaneous environment or immune response, leading to skin conditions such as demodicosis (demodectic or red mange), seborrheic dermatitis, and potentially rosacea (*Sastre et al., 2016*; *Jańczak et al., 2017*; *Forton & De Maertelaer, 2021*). *Demodex* mites contain lipase enzymes and the hydrolysis of sebum triglycerides releases fatty acids with irritant properties (*Jimenez-Acosta, Planas & Penneys, 1989*). Human patients with demodicosis have altered sebum composition, although it is unclear if this is a cause or consequence of the disease (*DemIrdağ et al., 2016*). Demodicosis can result in severe alopecia (*De Bosschere et al., 2007*; *Barlow & Wood, 2011*). Demodicosis is well known in humans, cats, and dogs but is generally considered rare in other domestic species, although local outbreaks occur (*Mauldin &*

*Peters-Kennedy, 2015*; *Nutting et al., 1975*). It has also been reported in a variety of captive and free-ranging wild mammals (*Sastre et al., 2016*; *Bianco et al., 2019*; *Carpenter, Freeny & Patton, 1972*; *Dräger & Paine, 1980*; *Forrester, Spalding & Wooding, 1993*; *Gentes, Proctor & Wobeser, 2007*; *James & Raphael, 2000*; *Salvadori et al., 2016*; *Nutting & Dailey, 1980*; *Takle et al., 2010*; *Wolhuter et al., 2009*; *Pence, Custer & Carley, 1981*; *Javeed et al., 2021*; *Nemeth et al., 2014*; *Barlow & Wood, 2011*; *De Bosschere et al., 2007*).

Ticks and mites, such as *Dermacentor* spp. and *Sarcoptes* spp., are attracted to specific components of skin lipids which may partially explain differences in occurrence among host species and body parts (*Arlian & Vyszenski-Moher, 1995*). Variations in the composition of skin secretions may also play a role in the attractiveness of hosts to tsetse flies (*Gikonyo et al., 2002*) and mosquitoes (*Obaldia et al., 2022*) that has implications for trypanosome and malaria parasite transmission. Sebaceous gland hyperplasia and seborrhoea (excessively oily skin) are some of the symptoms of sarcoptic mange caused by the mite *Sarcoptes scabiei* (*Bornstein, Zakrisson & Thebo, 1995*; *Oleaga et al., 2012*). Sarcoptic mange is a skin disease that affects a variety of wild mammals globally and is a threat to wildlife conservation (*Escobar et al., 2021*)

## Cancer

Skin tumors can develop in sebaceous glands, and are sometimes associated with papillomaviruses (*Sundberg et al., 1988*; *Casanova et al., 2017*). Sebaceous gland adenoma and carcinoma have been documented in a variety of captive wild mammals, but only once in a free-ranging individual (Baird's Tapir, *Tapirus bairdii*) (*Hubbard, Schmidt & Fletcher, 1983*; *Sundberg et al., 1988*; *Canfield, Hartley & Reddacliff, 1990*; *Obendorf, 1993*; *Owston, Ramsay & Rotstein, 2008*; *Bharathidasan et al., 2014*; *Majie et al., 2014*; *Matute et al., 2014*; *Arguedas, Guevara-Soto & Rojas-Jiménez, 2019*; *Kloft, Ramsay & Sula, 2019*). The prevalence, pathogenesis, and population effects of these tumors in wild mammals is unknown. It is also unclear whether sebum plays a role in cancer development.

## FACTORS AFFECTING SEBUM COMPOSITION AND QUANTITY AMONG MAMMALS

Sebum composition has only been described in 29 live, wild mammalian species (Table 1). Lipids from the fur of dead mammals (roadkill and skins in collections) are characterized in 22 additional wild mammalian species (*Lindholm et al., 1981*), but sebum composition may change after death, decomposition, or taxidermic preparation. The reported composition of skin lipids varies depending on the selected method of sampling and analysis (*Pappas, 2009*). This complicates comparisons among studies as different methods were used to target different classes of lipids.

Each mammal species characterized to date produces sebum of a unique composition (*Lindholm et al., 1981*; *Stewart & Downing, 1991*). A variety of factors may affect sebum quantity and composition among and within individuals of a species including hormones, season, skin pH, diet, age, and sex. Factors such as diet may also partially explain variation in sebum composition among species, but it is unknown why sebum composition is species-specific in all taxa characterized to date, and what mechanisms drive this variation.
## Phylogeny

There are similarities in sebum composition within some families and genera, such as within Canidae (similarities in diesters and cholesteryl esters) and among *Equus* spp. (similarities in wax diesters, lactones, and cholesteryl esters) (*Lindholm et al., 1981*; *Stewart & Downing, 1991*). Lipid composition can also be quite different within families, such as Sciuridae (differences in the presence of triolein and stearyl oleate) and Mustelidae (differences in the presence of triolein, stearyl oleate, cholesteryl oleate, and squalene) (*Lindholm et al., 1981*). There are large differences in sebum composition among families as different classes of lipids are present (*Lindholm et al., 1981*). These observations suggest phylogeny may partially explain some patterns in sebum composition, but further studies on a wider range of taxa using identical methods are required to resolve phylogenic patterns.

## Ecological associations

Ecological associations may be another factor influencing sebum composition. Several authors have noted the sebum of some aquatic or semi-aquatic mammals has large amounts of squalene, such as sea otters (*Enhydra lutris*), otters (*Lutra canadensis*), beavers (*Castor canadensis*), and sea lions (*Zalophus californianus*), as well as species in 'damp' environments such as eastern moles (*Scalopus aquaticus*) and kinkajous (*Potos flavus*; lives in rainforest) (*Downing & Stewart, 1987*; *Lindholm & Downing, 1980*; *Williams et al., 1992*; *Davis et al., 1988*; *Stewart & Downing, 1991*; *Wang et al., 2018*). Based on this observation, *Wang et al. (2018)* suggest squalene performs a function specific to mammals whose surface is often wet, yet squalene also makes up a large proportion of human sebum and is found in several species of bats (*Pannkuk et al., 2012*; *Pannkuk et al., 2013*; *Smith & Thiboutot, 2008*). Squalene is common in scent gland secretions of many land-dwelling mammals, such as pandas, peccaries, lemurs, and bats, and serves as a fixative to further extend the life of volatile compounds (*Dingzhen et al., 2006*; *Scordato, Dubay & Drea, 2007*; *Waterhouse et al., 1996*; *Wood et al., 2005a*; *Wood et al., 2005b*). Squalene is synthesized in all types of cells because it is a key intermediate in the formation of eukaryotic sterols, and is also found in prokaryotes (*Spanova & Daum, 2011*). Convincing evidence regarding ecological patterns in sebum composition awaits the characterization of a greater variety of mammal species.

## Body part

Sebum composition also varies within individuals, such as among body parts. Sebum on the surface of bat wings has more free fatty acids and sterol/wax esters than hair or wing epidermal tissue (*Pannkuk et al., 2012*). Lipid quantity varies among body parts in California sea otters (*Enhydra lutris*) as the skin had greater abundance of lipids than the fur, and the lower back had more lipids compared to other body parts such as the head (*Williams et al., 1992*). Lipid composition on hair varies among individuals and body parts in white-tail deer (*Odocoileus virginianus*) in terms of the quantity and occurrence of specific compounds such as decane and alkanes (*Gassett, Wiesler & Baker, 1997*). This variation may reflect different requirements among body parts in terms of sebum functionality. It may also contribute to differences among body parts in disease occurrence such as *Malassezia*- and ectoparasite-associated skin diseases as discussed in the previous section.

## Hormones

In humans and laboratory mammals androgenic hormones cause an increase in sebaceous gland size by stimulating both the rate of cell division and the rate of lipid accumulation (*Thody & Shuster, 1989*; *Stewart & Downing, 1991*; *Zouboulis, 2004*; *Makrantonaki, Ganceviciene & Zouboulis, 2011*). The increase in androgen levels at puberty in humans causes a large increase in the rate of sebum secretion and also changes lipid composition (*Stewart & Downing, 1991*; *Zouboulis, 2004*; *Makrantonaki, Ganceviciene & Zouboulis, 2011*). In contrast, estrogens tend to inhibit sebaceous gland activity and decrease gland size (*Thody & Shuster, 1989*; *Zouboulis et al., 2016*). The composition and quantity of human sebum varies with age (decreases with age) and sex, but there are also marked differences among individuals (*Thody & Shuster, 1989*; *Zouboulis et al., 2016*). In laboratory rats, sex and age-related differences in lipid composition are larger than differences in fur collected from various body regions within an individual, and much larger than inter-animal differences in age and sex-matched specimens (*Khandelwal et al., 2014*). There are several reviews that summarize the effect of various hormones on sebaceous glands in humans and laboratory mice and rats (*Thody & Shuster, 1989*; *Zouboulis et al., 2016*; *Smith & Thiboutot, 2008*). Given these patterns, hormones likely also have a major influence on sebum production and composition in wild mammals. For instance, scent glands are enlarged and more active in males of multiple wild mammalian species during breeding season when testosterone levels are high and oily skin secretions are visible on body parts used for scent marking (*Albone, 1984*; *Bakshi, 2010*; *Blank, Ruckstuhl & Yang, 2014*; *Buesching, Newman & Macdonald, 2002*; *Buesching, Waterhouse & Macdonald, 2002*; *Clarke & Frearson, 1972*; *Hardy et al., 1991*; *Kennaugh, Chapman & Chapman, 1977*; *Khazanehdari, Buglass & Waterhouse, 1996*; *Martín et al., 2014*; *Nassar et al., 2008*; *Pandey & Dominic, 1987*; *Pinter, 1985*; *Quay, 1953*; *Quay & Muller-Schwarze, 1970*; *Rasmussen, 1988*; *Stoddart & Bradley, 1991*; *Tomiyasu et al., 2018*; *Wood et al., 2005a*; *Wood et al., 2005b*). This pattern also occurs in scent glands of some domesticated and laboratory animals (*Thiessen, 1968*; *Jenkinson, Blackburn & Proudfoot, 1967*; *Ebling, 1977*). Injecting testosterone into females and castrates promotes the development of glands and secretions similar to mature males (*Mitchell, 1965*; *Clarke & Frearson, 1972*; *Stoddart, 1972*; *Jannett, 1975*; *Balakrishnan & Alexander, 1980*; *Pinter, 1985*; *Stoddart & Bradley, 1991*; *Iburg, Arnbjerg & Rueløkke, 2013*). Injecting progesterone into castrates can also increase the size and secretion rate of scent glands and increase the frequency of scent marking behavior (*Balakrishnan, Shelly & Alexander, 1984*). These hormonal patterns are not uniformly observed in wild mammalian species, such as kangaroo rats (*Dipodomys* spp.) (*Quay, 1953*; *Randall, 1986*). The effect of hormones on sebum quantity and composition between sexes and among age classes partially explains patterns of skin disease occurrence as discussed in the previous section.

## Season

Sebum composition and quantity varies seasonally in multiple species. For example, the amounts of myristic, stearic, and linoleic acids on the wing of *Myotis lucifugus* decreases over the hibernation season while pentadecanoic, palmitoleic, and oleic acid

levels increase (*Frank et al., 2016*; *Ingala et al., 2017*). Sebum composition also changes in porcupines (*Erethizon dorsatum*) as the free fatty acid portion of lipids coating the quills is higher in summer compared to winter (*Roze, Locke & Vatakis, 1990*). Sebum quantity is generally higher in summer than winter. For instance, sebaceous glands in moose (*Alces alces*) skin that are not part of specialized scent glands are reduced in winter and well developed in summer (*Sokolov & Chernova, 1987*). Similarly, sebum output in domesticated cattle is lower in winter compared to summer (*Smith & Jenkinson, 1975*) that may be caused by temperature differences. Sebum composition varied among domesticated cattle experimentally exposed to different temperatures (24 °C, 32 °C, 38 °C) over two weeks (*O'Kelly & Reich, 1982*). The amount of esterified fatty acids excreted in triglycerides decreased, while the amount excreted in wax esters increased with rising body temperature in the Brahman cattle breed, but not in the British breed (*O'Kelly & Reich, 1982*). Knowledge gaps remain regarding environmental effects on sebum composition and quantity and how these changes affect sebum function and disease susceptibility.

## Skin pH

Cutaneous pH can affect sebum composition in humans and laboratory mammals, and this may also apply to wild mammals. Some free fatty acids (a component of sebum) are generated within skin from phospholipids by secretory phospholipase $A_2$, and this enzyme is inactivated at alkaline pH (>7 pH), partially due to the activation of serine proteases (*Fluhr et al., 2004*; *Behne et al., 2002*). Acidic pH is also important for direct influence of lipid–lipid interactions in the lamellar bilayers of the permeability barrier (*Bouwstra et al., 1999*). Stratum corneum neutralization reduces competence of permeability barrier lipids (*Mauro et al., 1998*; *Hachem et al., 2003*). Sebum quantity and skin pH are inversely correlated in humans (*Wan et al., 2014*).

## Diet

Sebum composition is affected by diet (*Melnik, 2015*; *Lovászi et al., 2018*). Sebocytes synthesize all lipid classes present in sebum, but can also take up preformed lipids or remodel lipids from the bloodstream (*Zouboulis et al., 2016*). Severe caloric restriction or fasting in humans decreases sebum quantity and changes skin surface lipid composition as triglyceride and wax ester secretion is reduced (*Downing, Strauss & Pochi, 1972*). Young pigs fed a diet deficient in essential fatty acids develop altered skin lipid composition accompanied by scaly skin and greatly increased trans-epidermal water loss compared to pigs fed a regular diet (*Melton et al., 1987*). Dogs fed diets deficient in essential fatty acids develop seborrhoea, while supplementing their diet with sunflower oil or olive oil changed skin lipid composition and ameliorated symptoms (*Campbell & Dorn, 1992*; *Campbell, Uhland & Dorn (1992)*). Variation in sebum composition among bat species may be partially due to diet (*Pannkuk et al., 2012*; *Ingala et al., 2017*; *Frank et al., 2016*).

## CONCLUSIONS AND DIRECTIONS FOR FUTURE RESEARCH

Sebum is a physical and chemical barrier, and is important in thermoregulation, preventing water loss, maintaining the skin microbiome and healthy hair/fur, and protecting against

pathogens and abiotic stressors. Most research on sebum function and role in disease has been conducted on humans and laboratory/domestic mammals, but it is unclear how much these findings apply to wild mammals given species-specific differences in sebum composition. For example, inter-species comparisons are problematic in acne research as traditional laboratory mammals do not normally develop acne and have different sebum compositions from humans (*Schneider & Zouboulis, 2018*). The lack of transferability among species with regards to research on sebum function has been partially addressed through the use of genetically modified laboratory mammals and human sebocytes *in vitro* (*Schneider & Zouboulis, 2018*). These techniques may also facilitate laboratory studies on sebum function in wild mammals. Baseline data on normal sebum composition in uncharacterized mammal species may provide further insight on the biological roles of sebum and why sebum composition is species-specific in all taxa characterized to date. Further studies on the skin properties of mammals that lack sebaceous glands may provide insights into mechanisms that replace sebum functions when sebaceous glands are absent.

Skin is an effective physical and chemical barrier to pathogens and often skin disease only results when these properties are compromised by wounds, environmental factors (such as persistent wetting), or other diseases. Some ectoparasites, fungi, and bacteria on the skin only become pathogenic when the skin environment changes, such as disruptions of the protective lipid layer (over- or under-production of sebum), immune system, skin pH, or cutaneous microbiome. Infections, temperature, moisture, pollutants, U.V. radiation, and chemicals may change sebum quantity or composition that can subsequently compromise sebum functions and potentially lead to disease. Differences in sebum composition among species may help explain species-specific differences in disease susceptibility, since composition may impact sebum function and certain lipid components may have more effective antimicrobial functions against specific pathogens compared to other lipid components. A promising area of research is the effect of microbes on skin lipid composition, and vice versa, and how those effects contribute to skin defense against pathogen establishment and disease progression. Elucidating which microbes on the skin are important in generating free fatty acids or other lipids that prevent the establishment or growth of pathogens may facilitate biocontrol treatments for skin diseases. Determining which components of sebum different microbial species use for nutrition may provide insight into variations in the skin microbiome among and within individuals and species, given the wide variation in sebum composition. Standardized methods for testing microbial growth on different lipid components *in vitro* need to be developed and widely adopted.

Sebum quantity and composition varies with season, sex, age, and body part in some wild mammals. Based on research in domestic mammals, diet, skin pH, and hormones are likely also important in wild mammals, but have yet to be studied. Elucidating how these factors affect sebum quantity and composition in multiple taxa may provide insight into skin disease susceptibility that can also vary with these factors. Metadata such as sex, age, months samples collected, body part sampled, and captivity status should be routinely reported even if studies are not explicitly testing these variables. Currently, these factors are inconsistently reported which complicates comparisons among studies (Table 1). More data on sebum composition across taxa could clarify the roles of ecological factors and

phylogeny in shaping sebum quantity and composition. Currently it is not possible to assess patterns across taxa because of unstandardized methodology and the limited number of species that have been characterized.

It can be difficult to differentiate between epidermal and sebaceous lipids when studying skin surface lipids of mammals, especially outside laboratory settings. Epidermal lipids and sebum interact in ways that affect lipid composition and function. Additional research and metanalysis of existing studies on epidermal lipids and skin surface lipids in mammals may provide additional insights into functions performed by these lipids in maintaining skin health and preventing disease.

Additional studies are needed to further assess and clarify the contribution of sebaceous glands to skin maintenance and defense, particularly as new wildlife skin diseases are discovered. Such studies may uncover new therapeutic strategies and management options for mitigating skin diseases in wild mammals, which is increasingly important for species of conservation concern, whether wild or part of captive breeding programs. Although disease is a normal feature of the life of wild animals and management actions may not always be desirable, disease management can be viewed as an attempt to mitigate human actions that can cause or exacerbate diseases in wild populations (*Wobeser, 2002*). Multiple studies have reviewed the strategies and difficulties of managing disease in free-ranging wildlife (*Wobeser, 2002*; *Rowe, Whiteley & Carver, 2019*; *Lambert et al., 2021*). Understanding sebum function may lead to the development of drugs, topical products, or habitat modifications to mitigate disease occurrence and progression.

## ACKNOWLEDGEMENTS

Thank you to Dr. Donald McAlpine for helpful discussions and comments on the draft.

### Funding
The authors received no funding for this work.

### Competing Interests
The authors declare there are no competing interests.

### Author Contributions
- Karen Vanderwolf conceived and designed the experiments, performed the experiments, analyzed the data, prepared figures and/or tables, authored or reviewed drafts of the article, and approved the final draft.
- Christopher Kyle conceived and designed the experiments, authored or reviewed drafts of the article, and approved the final draft.
- Christina Davy conceived and designed the experiments, prepared figures and/or tables, authored or reviewed drafts of the article, and approved the final draft.

## Data Availability

This is a literature review.

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
