# Peer review of "A review of sebum in mammals in relation to skin diseases, skin function, and the skin microbiome"

_PeerJ, doi:10.7717/peerj.16680_

## Round 0.1 · original submission · Minor Revisions

We have been fortunate to receive four reviews for your manuscript. All reviewers agreed that your review paper was interesting and that it represented an important contribution to the literature in this field. Most comments provided by the reviewers are aimed at improving the flow and organization of your review.

In particular, the introduction - and some of the other sections - needs to be streamlined and more focused on wild mammals – see suggestions made by reviewer 2 and 4.

Figure 1 visual design could also be slightly improved.

Reviewer 1 ·

Basic reporting

This manuscript is a review of the functions of sebum with a focus on wildlife
populations and the roles it plays in various skin diseases and microbiomes. Overall, the paper is
well researched and well written. An adequate number of citations have been provided. A
synopsis table is extremely helpful with these review articles and one has been provided. I think this is a novel review that has been missing from the literature and will make a useful resource for people getting into research into this field.

Experimental design

no comment

Validity of the findings

no comment

Additional comments

no comment

Reviewer 2 ·

Basic reporting

This review by Vanderwolf et al. is of broad and cross-disciplinary interest, as it contains a very nicely summarized, yet detailed overview of sebum composition and function in wild animals, as well as a comparison to what is currently known in domestic animals and humans. The review is very comprehensive and will be of value to researchers studying animal and human disease, the microbiome, skin physiology, and more. Considering that reviews covering similar subjects are over 30 years old or are generally human-centered, an updated review on sebum composition and function in mammals is warranted. The authors also include the perspective of disease in relation to sebum composition and function in mammals, which is a valuable addition and one that has not been reviewed recently.

Experimental design

The authors do an excellent job of bringing together a large body of literature and digesting it into logical sections for the reader. Furthermore, the survey methodology is comprehensive and unbiased, and it is made clear what will and will not be addressed in the review. Overall, the review is organized and flows nicely. Suggestions for re-organizing or shortening certain sections can be found in detail in Additional Comments.

Validity of the findings

The argument set forth in the introduction centers around the idea that sebum plays an important role in both the skin microbiome and disease progression in humans and domestic animals, therefore sebum is also likely to play a role in these factors in wild mammals as well. Using the available literature, the authors provide a wealth of information to provide evidence for this argument. In addition, based on the current knowledge of sebum in wild mammals, the authors provide avenues for further research and address unresolved questions.

Additional comments

Additional comments and suggestions are listed below:

1. The writing is excellent! I found the review very cohesive, logical, and easy to read.

2. The introduction is a bit long. Can this be focused a bit more? What is the main information about sebum composition and function needed to better understand the body of the review? Some paragraphs in the intro seem like they could fit better into the body. For example, the paragraphs that discuss the microbiome.

3. In general, there was a lot included about sebum in humans (Malassezia sections, P. acnes sections, etc.). Because this review is focused on wild mammals, some of these human-focused sections could be shortened. Although it is understandable to include human-related information if the majority of what we know is from humans, maybe just clarify to the reader why there will be a focus on humans for a particular topic.

4. Similarly, it would be worth stating in the abstract that there is also knowledge synthesized on sebum function in humans and domestic animals, as well as wild mammals. The abstract reads as if only information about wild mammals will be discussed, but this is not the case.

5. The usage of “microbiomes” as plural throughout the review feels awkward, as “microbiome” is more generally used as a singular noun (“the microbiome”). In place, “microbial communities” is a common phrase that could be used if the plural version is preferred.

6. It would make more sense to introduce what sebaceous glands are (paragraph beginning line 170) first before discussing that sebaceous glands are absent in some species (paragraph starting line 158). I would recommend moving this paragraph to the end of the “Sebaceous Gland Occurrence and Anatomy” section.

7. The last paragraph of the introduction states that scent marking behavior will not be discussed. Does this entail any information about scent glands, or just the behavior itself? Because there is some reference to scent marking behavior (lines 708-710).

8. Similar to the above point, the section on scent glands (lines 198 - 229) feels a bit out of place and lengthy. This paragraph could be shortened significantly to focus the “Sebaceous Gland Occurrence and Anatomy” section.

9. The first three paragraphs of the “1. Functions of Sebum” section could be tightened (only a short paragraph or two), similar to the introductory paragraphs for the 2nd and 3rd main sections. Some of the information within this introductory paragraph could possibly be re-worked into the actual body paragraphs of the section.

10. Section 1.1 Skin barrier function: Lines 268 - 273 discuss epidermal lipids, however lines 196-197 state that studies on epidermal lipids will not be included in the review. It could either make sense to remove one or the other.

11. It could be valuable to include the recent paper describing the effect of defective lipid production in mice and the effect on the skin microbiota to add to the references found in lines 276-278: https://onlinelibrary.wiley.com/doi/10.1111/exd.14676

12. Line 318: Define “pelage”.

13. Lines 348 - 357: These sentences are focused more on sebum dysregulation as a result of UV stress rather than the function of sebum in UV stress. Consider moving these references to another section.

14. Lines 360 - 362: Consider removing this example about whales if other species that lack sebaceous glands do not have skin lesions caused by UV radiation, since it seems like sebum absence may not be the cause of this observation.

15. Change instances of Propionibacterium to Cutibacterium - There was a taxonomy change a few years ago, so P. acnes is now known as Cutibacterium acnes.

16. Line 398 - C. acnes, under normal circumstances, is found in the healthy skin microbiome, so consider removing “like P. acnes”. Also, consider rewording this sentence to say that free fatty acids may provide anti-bac activity against bacteria, rather than skin bacteria, since inhibition of the skin microbiota is usually not desirable.

17. Line 453: Introduce the animals that D. congolensis affects first perhaps before introducing the bacteria.

18. In the section about Malassezia, make sure to clarify when discussing human vs animal-associated Malassezia, such as the paragraph starting with line 528

19. Re-write sentence in lines 696-699 - was confusing to read

20. Section 3.5 Season - This section could benefit from some transition words/phrases. At the moment it reads like fact being listed after fact and is missing cohesion/flow between the various examples listed.

21. Lines 769-770: There is a typo in this sentence.

22. Line 773: What is meant by intrinsic interest?

23. Figure 1: Aligning the figure a bit more would help to make the figure seem more cohesive and clear (ie, Line up “Sebum Function”, “Disease Associations”, and “Sebum Composition” and have the subsection text also aligned in some way). Also, Capitalize the first word of the seb-sections for consistency.

24. Figure 3: Also capitalize first word of bullet points. Change the icon of the mosquito to some sort of microbe to make it clear that the pathogen is a microorganism rather than the mosquito itself.

Reviewer 3 ·

Basic reporting

o The review is well-written, with clear subheadings that neatly lay out the major topics to be discussed. Each subsection is clearly written, self-contained, and overall flows well into the next subtopic. I think this review is distinct from other papers on mammalian skin lipids and fits the scope of PeerJ well. It is appropriate for a general readership.

Experimental design

o Collection of articles appears robust and sensible, with clear explanations for how and why studies were included (or not). Probably the most comprehensive view of mammalian sebum I have ever seen with more than 250 references. All sources are adequately cited.

Validity of the findings

o As this is a review, the findings are more like hypothesis generated but seem interesting and follow logically from the summaries of published works. It is clear that there is still much to learn about sebum in wild mammals and this review frames many of the potentially testable hypotheses quite well.

Additional comments

o I really appreciate Figure 1 as an overarching concept figure. I wonder if the authors could clean up the aesthetics a bit though? Maybe by creating three different color blocks (as a background) for Sebum Function, Disease Associations, and Sebum Composition? I would also reorder these as function, composition, and disease association (at the bottom). I think that makes more intuitive sense.
o Line 605, just a brief note here would be good to make the reader aware that different attractiveness to tsetse flies / mosquitoes could have implications for trypanosome and malaria parasite transmission. It is obvious to those of us who work with diseases and vectors but not necessarily to a general readership like that of PeerJ.
o Line 638 (3.1 Phylogeny); I think that to say phylogeny partly explains patterns in sebum composition necessitates some kind of actual test of phylogenetic signal. In other words, a calculation of Pagel’s lambda or something. Without this evidence, it is difficult to say whether differences are idiosyncratic by species (each taxon has a unique sebum composition) or that sebum is more similar in closely related species. A phylogenetic explanation implies the later, but not the former. I understand that such a test may not be possible (depending on the data available in the referenced studies). If not, the text should be edited to reflect this uncertainty.

·

Basic reporting

- The nature of this review is unique, as it summarizes information from a topic that is quite narrow in focus (sebum) and couches it in the context of its role in a broader topic (sebum's role in disease). In that context, it fulfills two of PeerJ's stated standards by building out a cross-disciplinary study that can be useful to a large audience and providing new context for a review of this nature. I am also not aware of a review of this nature existing in the literature, which provides good justification.

- See comments below. The introduction provides a summary of what's presented but could be steered toward the author's stated goal of using this information to inform conservation and management practices. The introduction as written feels more like a repetition of the body of the paper and in that context it implies the goal of the paper is to provide a summary rather than aim the coalescence of knowledge toward an application.

Experimental design

- See comments below. While it could be difficult, there is room in this review to explore the role of sebum vs epidermal lipids vs skin surface lipids. The authors state they purposely aimed their work toward sebum, which makes sense for a lot of reasons. But potentially a deeper exploration of the way sebum and epidermal lipids combine, react, and change would help cover some of that missing context.

- References seem good and consistent, which is surprising and admirable with so many references.

- Logically organized based on the current manifestation of the paper.

Validity of the findings

- See comments below. The theme of skin diseases is carried throughout the paper. However, if desired, the themes of conservation and management are less pervasive.

- The conclusion follows the organization of the previous sections with a brief restatement of the themes and an exploration of additional work that could be helpful. It could be useful to reorganize the conclusion to avoid it feeling repetitive. Rather than a series of short paragraphs, instead organize them into comprehensive summaries of the entire topic in the themes that follow figure 1 (three paragraphs?) and then provide one, broad stroke paragraph that explores the need and benefit of further studies. This strikes me as a better way to wrap things up without losing conclusions that are spread across several paragraphs.

Additional comments

Hi Karen et al,

I want to start my review by stating that this effort is long overdue. This is a complex topic that is largely populated by medical and veterinary research, which is difficult to translate to wildlife given the lack of focused study on the ecological roles of skin surface lipids. Hopefully there will be renewed interest in this field, especially when framed in a wildlife diseases context.

When I started this review, my initial concern was whether there exists enough information on skin surface lipids in wildlife to build a substantive literature review. I’m happy to see that there is some good literature on the subject to the point that there is some difficulty in parsing the difference between your target literature (wild mammals) and other stuff (medical journals, laboratory studies, etc.). However, one issue that is not reconciled in this effort is the difference between the title of the paper, “A review of sebum composition and function in wild mammals” and the information presented in the review. You make a good point to state that there are only a few examples of these studies in wild mammals, but the majority of the review focuses on humans, laboratory studies, and domesticated animals, which are groups that you stated were excluded in the methods. Maybe this is simply a wording issue, but the point remains that the title’s spin toward “wild mammals” doesn’t have a lot of follow through in the body. Again, likely because this is difficult to do given the lack of focused study in this field, and I appreciate how you work to steer your findings toward wild systems given the opportunity, and I think that’s the best approach. First introduce the concept through what’s known in the lab and then expand it to the wild. Perhaps its more about being explicit in your set up of that strategy so your reader doesn’t come in expecting the entire review to focus on wild mammals.

Somewhere in the text (L197) you state that there is some difficulty in parsing studies that do/do not clarify whether they are discussing sebum vs epidermal lipids. This is a genuine challenge and I think another issue that points to reframing the review slightly. Perhaps it is more appropriate to refer to “skin surface lipids” rather than sebum explicitly. Your statement is that studies on epidermal lipids are not included in this review but in fact several of these studies almost certainly include a combination of epidermal lipids and sebum in the spirit of skin surface lipids (Pannkuk et al 2014, for example). I think this point needs more clarification and perhaps and extra paragraph or section that more explicitly explains the interactions between epidermal lipids and sebum such that you have as much coverage as you need to discuss relevant studies without clouding the function of sebum specifically. And this seems well justified because what is the biological relevance of studying sebum in isolation? It almost always is combined with other surface compounds, altered by the skin microbiome, and even oxidizes or otherwise spontaneously changes with exposure to the atmosphere. By limiting yourself only to sebum rather than skin surface lipids you likely steer yourself toward mostly human/laboratory models/domestic animals where researchers can be very explicit in their experiments with techniques and sampling.

The introduction is a broad overview of the body of the paper but as written it feels rather repetitive of the information provided below it, rather than a synthesis of the theme and the goal of the effort. Is the goal to summarize what’s available or make another point about the theme of disease resistance and application of the studies? I think the information is all there, just needs wordsmithing and reorganization to feel less like a string of citable facts. For example, in either the introduction or the conclusion, there could be a more explicit focus on application of this knowledge, which is a stated goal. A ctrl+F for “conservation” and “management” both return two occurrences in the main text, each time near the beginning or end of the paper. Stating how the information presented can be folded into conservation would help give the paper greater purpose and broaden the audience that may be interested in your conclusions. Do we have many examples? WNS of course, but surely there are other examples since you’ve worked the WNS angle into the paper a few times. Are there cases of managing mange for the benefit of wildlife populations? Devil facial tumors? You state that you don’t want to work treatments into the paper (and I agree) but understanding how those treatments can be effectively deployed is important if you want the paper to follow the theme of wild mammals, conservation, and management.

Overall, I think the style is good. The facts are presented in a logical order and do not overwhelm. The depth of information provided about WNS is much deeper than the rest of the literature summarized, possibly because of the intensity of study on the subject and the authors’ field of expertise. There could be space to expand on some of the wild systems presented, provided there is enough information to adequately develop the section.

I hope this review is helpful. You are welcome to contact me.

-Nate Fuller

---

## Round 0.2 · accepted · Accept

I have assessed the revisions made by the authors following the last round of reviews. The changes made are satisfactory and improved the flow of the manuscript. I am happy to recommend acceptance of this review which should be useful to many researchers in the field.